# “Working Together”: Perspectives of Healthcare Professionals in Providing Virtual Care to Youth with Chronic Pain during the COVID-19 Pandemic

**DOI:** 10.3390/ijerph20064757

**Published:** 2023-03-08

**Authors:** Danielle Ruskin, Julia Borsatto, Klaudia Szczech, Monique Tremblay, Lisa N. D’Alessandro, Giulia Mesaroli, Naiyi Sun, Catherine Munns, Jennifer Stinson

**Affiliations:** 1Department of Psychology, The Hospital for Sick Children, Toronto, ON M5G 1X8, Canada; 2Department of Anesthesia and Pain Medicine, The Hospital for Sick Children, Toronto, ON M5G 1X8, Canada; 3Department of Psychology, York University, Toronto, ON M3J 1P3, Canada; 4Lawrence S. Bloomberg Faculty of Nursing, University of Toronto, Toronto, ON M5T 1P8, Canada; 5Department of Rehabilitation Services, Hospital for Sick Children, Toronto, ON M5G 1X8, Canada; 6Department of Physical Therapy, University of Toronto, Toronto, ON M5G 1V7, Canada; 7Child Health Evaluative Sciences, Peter Gilgan Centre for Research and Learning, The Hospital for Sick Children, Toronto, ON M5G 0A4, Canada

**Keywords:** virtual care, pediatric chronic pain, multidisciplinary treatment, mixed methods analysis, healthcare professional perspectives

## Abstract

Background: The onset of the coronavirus disease in 2019 necessitated a rapid transition to virtual care for chronic pain treatment. Methods: A mixed methods design was implemented using qualitative interviews and quantitative satisfaction surveys. Interviews were conducted in February 2021 with a sample of healthcare professionals (HCPs; *n* = 6) who had provided multidisciplinary treatment (MDT) through an outpatient hospital pediatric chronic pain program. Satisfaction surveys were distributed to all MDT professionals employed by the clinic in April 2021 (*n* = 13 of 20 eligible; 65% response rate). Participants represented medicine, rehabilitation, and mental health professionals. Results: Analysis of interviews generated five themes: (1) adaptation to virtual care, (2) benefits of virtual care, (3) limitations of virtual care, (4) shifting stance on virtual care over time, and (5) considerations for implementing virtual care. The satisfaction survey data revealed that respondents were able to effectively provide appropriate diagnoses, recommendations, and/or care plans for pediatric chronic pain via virtual care (*n* = 12, 92.3%). Detailed survey responses are presented by discipline. Conclusions: This study provides a rich exploration of HCPs’ experiences in providing MDT for pediatric chronic pain within a virtual care model. The current results may contribute to the future development of guidelines for virtual care delivery with pediatric chronic pain populations.

## 1. Introduction

Chronic pain affects 20–35% of school-age children [1,2]. Those who present to chronic pain programs in tertiary care hospitals have pain that profoundly disrupts both the child’s life and their family life [3]. Common functional impairments in children with chronic pain include (a) school absenteeism (50–75% of children missing 1 day/month [4], 20% missing more than half of school days [5], and 17% of cases with complete withdrawal [6]), (b) physical limitations (e.g., reduced physical activities, poor endurance, and generalized deconditioning), [7] and (c) mental health sequelae and comorbidities, particularly anxiety and depression, which co-occur with pain in 21% to 26% of children [8]. The impacts on family life are profound, including financial strife due to parental days off work and limitations on family outings and activities [9].

Given that a latency of 6 months from the time of referral to treatment for chronic pain is associated with deterioration in health-related quality of life and psychological wellbeing, the importance of efficient and timely care for chronic pain cannot be overstated [10]. Therefore, in response to the pandemic from the coronavirus disease of 2019 (COVID-19), the delivery of care for youth living with chronic pain drastically shifted in an attempt to provide timely care. Beginning in March 2020, the world saw pediatric pain management clinics abruptly changing their delivery of care owing, in part, to social-distancing requirements. Depending on the location, some healthcare professionals (HCPs) were unable to see patients for varying periods of time because of clinic shutdowns or redeployment, whereas others were able to rapidly pivot to a virtual model of care delivery [11,12,13]. Fortunately, across Canada, all 13 pediatric pain clinic sites were able to successfully transition to virtual care, where the largest program in Canada (The Hospital for Sick Children) transitioned over the course of 2 weeks at the onset of the pandemic [11,14].

Further, given that the optimal paradigm for chronic pain management involves psychological, physical, and pharmacological approaches, the transition from in-person care to a full-scale virtual model was no easy feat [15,16]. Multidisciplinary treatment (MDT) teams consisting of physicians (e.g., anesthesiologist, pain doctor, or pediatrician), physical therapists, nurse practitioners, psychologists and/or psychiatrists, occupational therapists, and/or social workers began to rapidly navigate virtual technology to ensure that patients received safe and quality care [13,17]. While some HCPs had prior knowledge of and practice delivering occasional telemedicine appointments, very few had experience delivering full-time remote services (e.g., videoconference, telephone consultation) [12,18,19]. HCPs faced many administrative, logistical (i.e., infrastructure, setting up virtual care), and technological (i.e., internet connection, technological literacy) challenges, but they demonstrated resilience in meeting the demands of the unprecedented circumstances [13,17].

Now, nearly 3 years since the initial shift in modality of care, it is apparent that virtual healthcare has endured beyond needing to meet social-distancing requirements [20,21,22]. Virtual care has proven to be an advantageous method of delivery across populations and disciplines, with geographical and financial accessibility frequently touted as benefits [18,20]. However, its delivery also possesses a unique set of challenges. When considering specifically the delivery of virtual pediatric pain care, HCPs described the inability to perform in-person physical exams, the blunting of nonverbal and physical cues, and inequitable access to the internet and technological devices as major limitations. These findings come from an investigation by Killackey et al. [17], who conducted a qualitative analysis of various HCPs’ experiences in shifting to virtual pediatric pain care during the pandemic. The present study aims to build on these findings and enhance the understanding of providing this care in the context of an MDT team. Using mixed methodologies (semistructured interviews and self-reported satisfaction surveys), the primary aim of the current study was to elicit feedback from an MDT team of HCPs on their experiences assessing and treating pediatric chronic pain patients by utilizing virtual care during the second wave of the pandemic in Canada. A secondary study aim was to explore differences in experiences across disciplines in the multidisciplinary team (i.e., mental health, rehabilitation, and medicine).

## 2. Materials and Methods

### 2.1. Study Design

Collecting stakeholder feedback is essential to program evaluation and is best obtained by using combined quantitative and qualitative data to capture the richness and complexities of program implementation [23]; therefore, a mixed methods design was employed in the current study. Both qualitative (semistructured interview) and quantitative (self-report satisfaction survey) methods were used to identify and evaluate the experiences of HCPs delivering virtual MDT throughout the second wave of the COVID-19 pandemic in Canada (e.g., February to April 2021). Qualitative interview questions mirrored areas queried in the satisfaction survey but provided additional opportunity for richer feedback (see Section 2.6. for additional details).

### 2.2. Setting

The current study was conducted at a pediatric chronic pain clinic providing outpatient MDT pain assessment and management within a large urban tertiary care hospital. Ethics approval for this research study was obtained from the institution’s research ethics board (REB #1000071310).

### 2.3. Virtual Technology Platform

In this paper, “virtual MDT care” refers to psychology and physiotherapy assessment and treatment delivered using PHIPA (Personal Health Information Protection Act)-compliant Zoom for healthcare.

### 2.4. Implementation of Virtual Care Model

The chronic pain clinic at which this study took place successfully transitioned all clinic appointments (MDT intakes, assessments, and treatments) over a 2-week period to virtual care [11]. Patients referred to the clinic engaged in a 2-hour MDT intake visit (with a physician, psychologist, physiotherapist, and advanced practice nurse) composed of (1) reviewing medical and pain histories and current and prior functioning, (2) a physical exam, and (3) the disclosure of pain diagnoses and feedback as well as recommendations for treatment. This process, which previously occurred in person, was readily transferred to the virtual modality and was conducted in the same manner, albeit virtually. Following the MDT intake visit, separate psychology and physiotherapy assessments were completed. After the initial psychology and physiotherapy assessments, patients deemed appropriate for further treatment were then engaged in a block of treatment with the MDT.

Each treatment block consisted of four sessions of weekly psychology and physiotherapy care (one hour each) and a 1-hour pain education session delivered to patients and families, with adjunctive pharmacology and occupational therapy support as needed (see Figure 1). Additional complementary therapies were offered, including a virtually delivered group for youth on mindfulness-based pain management. Prior to the pandemic, the above treatments were delivered in person, aside from infrequent virtual care treatments for those patients living in rural and remote areas of Ontario, Canada.

The majority of psychology and physiotherapy assessments and treatment sessions had been virtually delivered over Zoom since the onset of the COVID-19 pandemic (88.8% of physiotherapy appointments and 96.4% of psychology appointments over the study period). For selected cases, our team provided in-person assessments where virtual care would not have been feasible—e.g., very young children (<5 years old), those with developmental considerations, children who required an in-person physical exam (e.g., those with neuropathic pain or complex regional pain syndrome and those who have not had a physical exam), or individuals who faced barriers to accessing virtual care (e.g., low computer literacy, appropriate bandwidth, and lacking access to technology).

### 2.5. Recruitment and Participants

Utilizing purposive sampling to ensure that each discipline that provides MDT in the institutions’ pediatric chronic pain team was represented, a portion of the clinic’s HCPs were invited to participate in semistructured interviews (SSIs) regarding their experience in delivering virtual care. It was determined a priori that a sample of six participants was adequate for producing cross-case generalities while still considering each participant as an individual identity [24]. The six participants for qualitative interviews were selected by using the following procedure: HCPs with full-time equivalent (FTE) positions were approached first because these individuals would have the maximum hours spent providing virtual care. If an HCP approached for the interview declined, then the next HCP in their discipline was invited to participate on the basis of FTE. Information on the purpose of the study, a consent statement, and invitations to participate were sent to HCPs by a remote member of the institution’s research team (K.S.) in February 2021.

To assess the extent to which the experiences described by the subset of HCPs generalized to the entire MDT team, a questionnaire examining HCP satisfaction with provision-of-pain assessments and treatments via the virtual modality was disseminated to all MDT professionals employed by the clinic. Information on the purpose of the surveys, as well as the access links, were emailed by K.S. in April 2021. Surveys were hosted in Redcap (Research Electronic Data Capture), a secure, web-based application designed to support data capture for research studies [25]. An implicit consent statement was included with the study purpose, and HCPs were assured that participation in the study was voluntary and would not impact their employment. Primary investigators were excluded from recruitment.

### 2.6. Procedure and Measures

SSIs were virtually conducted by using PHIPA-compliant Zoom throughout February 2021. The interviewer was a predoctoral psychology resident (M.T.) with training in qualitative interviewing. SSI guides (see Appendix A) were created by the authors to generate rich, in-depth discussion on HCPs’ experiences in providing virtual care. Informed by Turner [26], questions were open ended and single faceted, with prompts such as “tell me more about that” used to elicit detail and clarity, when necessary. Interviews were audio recorded and transcribed verbatim. Staff satisfaction survey content was informed by a complementary patient/caregiver satisfaction study [27] and was refined by using themes generated from the SSIs through an exploratory design procedure [28]. Questions probed various aspects of virtual care, including equipment and technical issues, communication and rapport, clinical modifications, and perceived self-efficacy of assessment. Respondents could also provide recommendations on the future of pediatric pain care delivery. Response options varied across questions and included text entry, true/false, and a 7-point Likert scale (1 = strongly disagree, 7 = strongly agree). See Appendix A for the complete satisfaction survey.

### 2.7. Data Analysis

#### 2.7.1. Qualitative Analysis

SSIs were transcribed verbatim and reviewed for accuracy. Following transcription, the data were coded by K.S. and J.B. in Dedoose [29], a SaaS coding software program that can be used to organize and assist with coding qualitative data. Taking a reflexive thematic analysis approach, coders met biweekly to refine themes and subthemes in an iterative and rigorous manner [30]. Disagreements were resolved by consultation with another author (D.R.) as needed. Following these meetings, a coding manual was generated in Dedoose, and an interrater reliability coding test was created. Specifically, one of the coders (K.S.) selected 92 excerpts from the coding project file and applied themes and subthemes to these excerpts. The unlabeled excerpts were then entered into a test template for the second coder (J.B.) to apply themes and subthemes to assess the degree of agreement between coders. An interrater reliability analysis using the Kappa statistic was performed to determine consistency among raters [31].

#### 2.7.2. Quantitative Analysis

Descriptive data were analyzed from the satisfaction surveys. For ease of interpretation, responses to items one to three and five to seven were merged into agree and disagree categories, respectively. Lastly, “neither agree nor disagree” responses remained as one response category.

## 3. Results

### 3.1. Semistructured Interviews

In total, six HCPs from a group of six eligible participated in the SSI (100% response rate). Five overarching themes, namely (1) adaptation to virtual care, (2) benefits of virtual care, (3) limitations of virtual care, (4) shifting stance on virtual care over time, and (5) considerations for implementing virtual care, as well as corresponding subthemes, were generated. They are described below. Disciplines have been attached to the quotes below in the following manner: medical professional includes physicians and nurses, rehabilitation professional includes physiotherapists and occupational therapists, and mental health professional includes psychologists and nurse practitioners (psychiatry). Please see Table 1 for a full list of the themes and subthemes generated from the interviews.

#### 3.1.1. Theme 1: Adaptation to Virtual Care

The first major theme examines participants’ observations of the shift from the in-person delivery to the virtual delivery of pain assessment and treatment. Three factors were echoed by multiple HCPs across disciplines, comprising three subthemes: (a) teamwork/togetherness, (b) virtual modality necessitating innovations in care delivery, and (c) ease of transfer to virtual care given the service model.

(1)Teamwork/togetherness—many HCPs described that the MDT team had to work closely with each other to navigate new technology and ensure treatment goals were being adequately addressed:

*“I think it was really great, everybody came together and we were developing these different protocols, you know, quite quickly… it was a steep learning curve but really great for everybody because we all put our heads together and were trying to deliver the best care to our patients and families.”* (HCP06, mental health professional)

Group cohesion as well as an openness to sharing knowledge with each other was essential in the shift to virtual delivery:

*“I think it came with a bit of a learning curve because there was no formal teaching on how to [transition to virtual care]. And so, we just kind of had to fly by the seat of our pants, but managed through sharing information and doing a bit of research.”* (HCP03, rehabilitation professional)

(2)Virtual modality necessitating innovations in care delivery—participants often needed to flexibly adapt and/or modify their assessments to adequately deliver service and address treatment goals in the virtual space, as shown in the following quote:

*“To do a handwriting assessment I’ll sometimes ask them to send me an actual video of them handwriting. It’s just easier than seeing it on the screen sometimes, they can get better angles with their phone.”* (HCP01, rehabilitation professional)

Further, clinicians in the mental health discipline also highlighted the use of the virtual modality as a form of treatment:

*“When they have social anxiety, they don’t want to be on the screen and seeing themselves. You can actually use that as a little bit of an intervention and exposure.”* (HCP06, mental health professional)

(3)Ease of transfer to virtual care given the service model—when considering the overall shift to virtual service delivery, participants echoed sentiments that the multifaceted delivery of pain care made this adaptation relatively seamless. One participant summarized this as follows:

*“I would say it’s the nature of chronic pain assessment and treatment that it lends itself to being more successful virtually, as opposed to primary care… a lot of it is not hands-on type of treatment. So, whereas assessment it’s helpful to get a bit of the hands-on, if possible, the treatment is mostly active therapy.”* (HCP03, rehabilitation professional)

This sentiment was expanded on by other HCPs, who emphasized that many aspects of pain treatment boded well in the virtual modality:

*“My treatment, delivery of pain education… [consists of] the debrief and giving of diagnoses and recommendations, providing prescriptions and so, I didn’t feel that there were barriers to [virtual care], no.”* (HCP05, medical professional)

#### 3.1.2. Theme 2: Benefits of Virtual Care

This theme was generated to reflect the common benefits of virtual care delivery described by HCPs. Many participants described virtual appointments as being more geographically and financially accessible (e.g., no parking fees, families not taking time off work). The related two subthemes are (a) convenience and comfort and (b) continuity of care.

(1)Convenience and comfort—the convenience of providing care through virtual appointments allowed for changes in some participants’ daily routines that decrease in logistical stress:

*“I don’t have to get up so early in the morning, I’m not racing to work, I’m not, [going to] the clinic, all the logistic things. So, I feel, personally, a little more refreshed.”* (HCP05, medical professional)

HCPs also described that seeing the patients in the comfort of the family’s home environment boded well for rapport and for treatment planning:

*“But the other benefit, too, is you actually get to see and get more information about what the patient and family’s home is like, and what the conditions are like, that you wouldn’t necessarily see in an outpatient visit.”* (HCP06, mental health professional)

(2)Continuity of care—the ability to provide pain care despite the restrictions related to the pandemic was another benefit of the virtual model of delivery described by many participants:

*“I think being able to continue to provide the care using the virtual technology allowed us to continue to support these kids who are really struggling with their pain, so in that way, it’s been positive, that we’ve been able to do that. And not shut down our clinic like some of the other programs had to do.”* (HCP02, medical professional)

HCPs were able to successfully navigate new technology to continue to meet treatment goals:

*“I’m really proud that our clinic really did a quick, kind of, pivot to virtual care, and we didn’t have huge gaps in providing care to our patients and families, and I think that’s a real strength.”* (HCP01, rehabilitation professional)

#### 3.1.3. Theme 3: Limitations of Virtual Care

Participants identified several limitations when providing virtual multidisciplinary treatment for pediatric chronic pain. The noted challenges included (a) limitations in observing nonverbal cues, (b) barriers to physical examination, (c) privacy concerns, (d) technology glitches, (e) virtual fatigue and engagement challenges, and (f) inequities in access.

(1)Limitations in observing nonverbal cues—participants discussed the limited nonverbal communication and feedback in virtual care, which had a reciprocal effect on both HCPs and patients:

*“When we’re in a room with the patient, I have the privilege of them looking at me, looking at others, picking up on nonverbal communication that is just not present in a video format. So, it’s not about comfort, it’s just about the fact that I don’t feel like I’m really getting the best experience with observing the patient, or them receiving any kind of feedback from me, even if it’s not verbal, just an acknowledgement when they’re sharing whatever it is. They don’t get the nods, and the encouraging words, because it’s much more scripted and formal.”* (HCP02, medical professional)

Further, restricted nonverbal cues in the virtual modality impacted clinical observation, as one participant described:

*“With [virtual] delivery, you can’t see if their leg is shaking under the table, or maybe they’re wringing their hands, or things like that. Or for depressive symptoms necessarily, their hygiene.”* (HCP06, mental health professional)

(2)Barriers to physical examinations—especially pertinent to multidisciplinary treatment for chronic pain, participants recognized the challenges of performing a physical examination as part of virtual care. One participant reported the following:

*“In terms of, let’s say, the initial assessment, which we have to do for the observation, like, very thorough examination of a joint to really determine whether there is anything else medical going on. So, not just gross motor movements. I remember a time where it was just really hard to see, like, is there swelling on the foot, is there a change in color? We have to go by the patient’s report, which isn’t ideal.”* (HCP03, rehabilitation professional)

Additionally, patients were at times brought to the clinic for in-person care in response to such barriers:

*“There was one time that I joined like a multi-d assessment with a physiotherapist, a psychologist, the whole team. And the child actually had a lump on his back that we couldn’t see through the video, just because of the quality. So, I mean that one, like, we ended up bringing him in eventually, but, yeah we couldn’t assess that properly through the multidisciplinary virtual assessment.”* (HCP04, mental health professional)

(3)Privacy concerns—participants across disciplines expressed concerns regarding privacy limitation associated with providing virtual care. It was noted that such challenges have the capacity to limit patient disclosure:

*“We have no idea who is sitting behind that kid, or outside of the camera. And I feel like sometimes the kids will just not open up and share things with us that they otherwise would’ve shared with us one on one. I can’t tell you how many times in in-person visits in the past, when we’ve had those teens on their own, some of the things they share with us, they would say “don’t tell my mum this, don’t tell my dad this” and they would disclose really important issues that have allowed us to, really, work that in with their care. And now, I just don’t feel like we can get that genuine feedback from the kids.”* (HCP02, medical professional)

At other times, privacy concerns completely prohibited aspects of virtual care:

*“There were a couple of situations where either the patient had come and was, kind of, logging in from, a non-private place and you could tell that they weren’t 100 percent comfortable. So, in that situation they actually declined the assessment, the physical part of the assessment.”* (HCP03, rehabilitation professional)

(4)Technology glitches—participants identified technology-specific challenges such as audio-video interruptions and bandwidth limitations, which occasionally impeded treatment:

*“If there were technical challenges and it kept on zoning out, or kind of, glitching, then I wasn’t able to complete the session.”* (HCP03, rehabilitation professional)

Participants also outlined that technological disruptions occurred on both HCP and patient ends:

*“Some of the families live so remote that they had very intermittent internet. Or even some of the staff were challenged where other family members in the home were also using the computer so it was affecting the bandwidth, so some people would cut out in the middle of an appointment, or their video would go off.”* (HCP02, medical professional)

(5)Virtual fatigue and engagement challenges—limited engagement during virtual appointments was noted as a challenge for participants. This was amplified when both patients and HCPs alike experienced screen fatigue from engaging in so many virtual interactions:

*“I think for a few cases it would definitely be beneficial for them to come in, engagement can be hard sometimes virtually, again because they’re always online. And I know that the kids get overwhelmed with all the virtual appointments, that’s not only with our clinic, a lot of our kids are complex, so they have so many virtual appointments. I get tired of looking at screens and I know the kids do.”* (HCP01, rehabilitation professional)

Participants also described limited engagement and rapport with caregivers during virtual appointments:

*“I found that I wasn’t able to necessarily engage as much with the parents. When they would come in to the hospital, the parent would always be with them, so we would do a recap at the end. At this point, especially the teenagers, who are really doing this whole online thing on their own, the parent wouldn’t be present or they’d be in a meeting, and I did lose some of that rapport with the parents.”* (HCP03, rehabilitation professional)

(6)Inequities in access—insufficient access to the resources required for effective virtual care, such as a home computer, stable internet, and a confidential environment, was reported as a limitation by participants:

*“I also think about families who don’t always have access to a stable internet connection, access to computers, access to a safe space.”* (HCP01, rehabilitation professional)

Another participant described erroneously assuming that all patients have access to the required resources:

*“Sometimes for families, they don’t have a confidential space, or they actually don’t have data on their phone, or regular internet access, so maybe they have to go and use a neighbor’s. We assume that everybody has internet access, they have good data plans, and we are assuming that, and they don’t necessarily have that.”* (HCP06, mental health professional)

#### 3.1.4. Theme 4: Shifting Stance on Virtual Care

A theme was constructed to describe how the evaluations and perceptions of virtual care from the perspective of HCPs, patients, and caregivers change over time. Participants’ statements were organized into two subthemes: (a) the change in HCPs opinions on virtual care and (b) the change in patient and caregiver perception of virtual care.

(1)The change in HCPs’ opinions on virtual care—participants reported an overall positive shift in opinion on providing virtual care, from the beginning of the COVID-19 pandemic onward:

*“For me personally, initially it was very stressful, and then it got much easier. To the point where I think it’s going very, very well overall, in terms of providing the care.”* (HCP02, medical professional)

It was also noted that although physical-distancing restrictions facilitated the mandatory shift to virtual care, this transition encouraged HCPs to view the potential of providing virtual care beyond the COVID-19 pandemic:

*“I think it’s been a good experience. It’s really broadened my view of providing telepsychology and virtual care, so I think it’s been a good experience.”* (HCP06, mental health professional)

(2)Change in patient and caregiver perception of virtual care—in contrast to the positive shift of opinion toward virtual care described by HCPs, participants reported a negative trend in patient and caregiver perceptions of virtual care, citing scheduling challenges as a main concern:

*“What has shifted, is that we’re getting more pushback from families about when they will schedule appointments with us because their kids are now in online school, and the kids are telling us they have pressure, they don’t want to miss class because of this need to keep up with the other students. Now, which I find interesting, because kids were missing almost a full day of school to come for their treatment prior to COVID, so I’m not quite sure what’s behind that. Again, I don’t know if it’s just that expectation, but I think, some of that positivity initially, I think is probably shifting.”* (HCP02, medical professional)

Another echoed this sentiment:

*“As things have progressed now, because it’s more informal, and maybe there’s an appearance that it’s easier to schedule, that often patients and families are wanting the virtual appointments to move around their schedule with school. So, they’ll be asking for evening appointments, or weekend appointments. Whereas, I mean, and we’re a pain clinic in a hospital, so I don’t know another ambulatory clinic like cardiology where you would have evening appointments or weekend appointments.”* (HCP06, mental health professional)

#### 3.1.5. Theme 5: Considerations for Implementing Virtual Care

The final theme addressed considerations and recommendations for implementing virtual multidisciplinary treatment for pediatric chronic pain. This theme included (a) the preference for a hybrid model and (b) recommendations for implementing virtual care.

(1)The preference for a hybrid model—HCPs across disciplines outlined a preference for hybrid care, which would incorporate both in-person care and virtual care as needed according to evidence-based practice and family needs:

*“I think we need to ensure that we are providing treatment that is in accordance with best evidence and that is in accord with the child and family’s preferences and values. And then I think that, what would be ideal, is to have a hybrid model. So that if the patients want to come in, that they see the people who you feel really need to see them.”* (HCP05, medical professional)

Participants also specified a preference that initial appointments for new patients be in person at the clinic, in order to build rapport and effectively obtain confidential information from patients in a secure space:

*“What I think would be amazing is if the new patients were coming in person so we could all get to know them, get to hear their story, pull those kids separately, get the information, you know, in confidence, which we would normally do. But then maybe, return appointments could be done in this type of format for convenience. But that way they’ve already met the team and they kind of know what it’s all about. So maybe a blended model of the two.”* (HCP02, medical professional)

(2)Recommendations for implementing virtual care—finally, participants outlined their recommendations and suggestions for adopting and optimizing virtual multidisciplinary care for pediatric chronic pain. It was noted that technological considerations must be factored in to achieve clear and uninterrupted virtual appointments and to enhance the virtual care experience:

*“I think people could do a better job of optimizing [the technology]. So, I think to have a crisp picture and really good sound is really important to reduce the impact of not being there in person.”* (HCP05, medical professional)

One participant suggested providing scheduling instructions for families to mitigate the abovementioned scheduling challenges when booking virtual appointments:

*“Giving [patients] the guidelines of when, you know, setting the appointments, not necessarily having so much flexibility, even though it appears that it’s not a big deal. If we made that a little bit more clear, maybe there would be less of those requests.”* (HCP03, rehabilitation professional)

Participants also commented on the pilot triage system that was developed by the clinic’s HCPs to schedule patients for virtual or in-person appointments:

*“We came up with criteria for triage to say, here are some specific populations that we would prefer come in person for their appointment, rather than doing this virtually. And that was through trial and error with specific diagnosis, and age group, and whatnot.”* (HCP02, medical professional)

### 3.2. Satisfaction Surveys

In total, 13 HCPs from a group of 20 eligible HCPs completed the satisfaction survey (65% response rate). Overall, 92.3% (*n* = 12) of respondents described virtual care as an effective treatment for pediatric chronic pain and agreed that virtually gathered information pertaining to a patient’s symptoms was sufficient to inform appropriate diagnoses, recommendations, and/or care plans. Rapport did not appear to be impacted by virtual care, all participants indicating an ability to develop a therapeutic relationship with the patient and caregiver.

Of the 13 participants, 5 HCPs were in medicine, 5 were in rehabilitation, and 3 were in mental health. When asked to consider ways that virtual care differed from in-person care, 40% (*n* = 4) of HCPs from the medicine and rehabilitation disciplines rated that the discussion of sensitive topics (e.g., self-injury) was more, or equally, challenging between modalities. HCPs in the mental health discipline did not endorse a difference. Similarly, when probed about the absence of physical contact in a virtual assessment, all HCPs in mental health (*n* = 3, 100%) agreed it was not a problem. However, 50% (*n* = 5) of professionals in the medicine and rehabilitation disciplines reported that the lack of in-person physical examination made virtual assessment more challenging. Notably, they shared sentiments that assessments of skin color, temperature, texture, range of motion, and muscle tension had to be modified (e.g., self-assessment, separate video submissions).

Regarding technological logistics, HCPs across disciplines (*n* = 10, 76.9%) described an overall positive experience, whereby they encountered few to no technical difficulties. When asked about their preferences for virtual use in the future, all HCPs participants agreed they will continue to use virtual care in some capacity. Participants expressed a preference for a hybrid model of delivery (*n* = 11, 84.6%), whereby some sessions are delivered in person and some virtually. Mental health professionals (*n* = 3, 100%) were more apt to remain virtual, even if in-person appointments were offered. Please see Table 2 for respondent characteristics. Table 3 outlines the survey responses where all HCPs (*n* = 13) were in agreement, and Table 4 provides differences observed across disciplines.

## 4. Discussion

The primary aim of the present study was to examine the experiences of health care professionals who delivered multidisciplinary pediatric chronic pain care throughout the second wave of the COVID-19 pandemic in Canada. A secondary aim was to explore differences in experiences across the disciplines in the multidisciplinary team (i.e., mental health, rehabilitation, and medicine). The current study’s qualitative and quantitative analysis supports the growing body of literature describing the overall benefits and challenges of providing virtual care [12,13,32,33,34], and it corroborates Killackey et al.’s [14] similar investigation into HCPs’ experiences in providing pediatric pain care during the COVID-19 pandemic. Further, this study provides a richer understanding of providing virtual care in the context of multidisciplinary pain management for pediatric chronic pain patients.

Specifically, the results from the descriptive quantitative analysis support virtual care as an effective modality to inform chronic pain diagnoses and recommendations in care for children living with chronic pain. This is in line with many studies across both adult and pediatric populations, suggesting that virtual care enabled HCPs to provide essential treatment and intervention throughout the pandemic [13,18]. Notwithstanding the benefits related to this continuity of care, the rapid uptake of virtual treatment and assessment highlighted the barriers of providing virtual care. Particularly, medicine and rehabilitation professionals indicated that the lack of hands-on physical examination made virtual assessment more challenging. These findings align with other studies, which described the challenges of performing physical exams and the absence of nonverbal cues as being HCP-reported limitations of virtual care [17,35]. Further examinations of HCPs’ experiences highlighted that professionals in the mental health field experienced increased comfort discussing sensitive topics (e.g., self-injury) virtually than those in medicine or rehabilitation. It is possible that because mental health professionals elicit sensitive information as a standard component of their work, this is an area of comfort. In addition, the institution where this study was conducted disseminated a clear, stepwise plan to hospital staff, with a tailored protocol for mental health professionals to address safety concerns divulged during a virtual appointment (E. Romanchych, personal communication, 20 April 2022). This may also have contributed to the increased comfort levels of mental health professionals in discussing sensitive topics.

Our qualitative analysis complements these findings and highlight five main themes related to HCPs’ experiences: (1) adaptation to virtual care (2) benefits of virtual care, (3) limitations of virtual care, (4) shifting stance on virtual care over time, (5) considerations for implementing virtual care.

A key finding in the first theme was the perception that a high level of cohesion and togetherness between the HCPs across disciplines aided in the shift to virtual care. This finding offers an opportunity to better understand characteristics described as common features found in high-functioning healthcare teams [36,37]. Moving forward, it may be relevant for leadership/management personnel to encourage team-bonding experiences to facilitate, maintain, and improve teamwork within a multidisciplinary team for the benefit of their patients [36]. In addition, our data highlighted the flexibility and creativity that HCPs demonstrated throughout the transition from in-person care to virtual care. Providers described requiring to flexibly adapt and/or modify their assessments to adequately deliver service and address treatment goals. For example, mental health professionals described using the virtual whiteboard function in lieu of in-person sketching that would occur during in-person sessions of cognitive behavioral frameworks, as well as incorporating breakout rooms in larger treatment groups to allow for smaller group work. Richardson and Kandu’s [13] paper outlined “harnessing of creativity and innovation” as a common theme in the pediatric pain world, noting HCPs’ uptake of tools such as mobile applications, social media, blogs, podcasts, and online support groups to improve patient outreach and treatment.

The second and third themes were developed to describe the benefits that HCPs believed virtual care offered (e.g., continuity of care, convenience/comfort) and its limitations to the provision of pediatric chronic pain (e.g., barriers to nonverbal observation and physical examination, privacy concerns, technology glitches, virtual fatigue/engagement challenges, and inequities in access). As described, these outcomes are consistent with many other studies exploring the delivery of pain care in the virtual space [12,13,17,33,34]. Notably, the virtual fatigue/engagement challenges identified by our participants appear to be all too common across healthcare professionals, patients, and families [19,38]. To reduce screen time throughout the traditional work/school week, HCPs in the present study noted an increase in patient/caregiver requests for afterhours appointments (e.g., weekend times). Moreover, they identified patient “Zoom fatigue” to be a precipitating factor to our fourth theme—shifting stances on virtual care.

Indeed, HCPs described observing two shifts in the perceptions of virtual care. First, they observed patients’ and caregivers’ initial enthusiasm for virtual care dissipate over time, citing virtual fatigue and scheduling challenges as the main factors driving this change. As previously noted, HCPs observed an increase in weekend or afterhours appointment requests and experienced perceived frustration from patients and families when these requests could not be met. Research has suggested that matching patients’ booking requests with providers’ availability is a major barrier when creating outpatient appointments and may contribute to a reduction in patients’ perceived value of care [39,40]. Recommendations to mitigate this perception include using clear communication while reviewing patient expectations and preferences and ensuring all staff are adequately trained in effectively scheduling clinical activities [39]. The second shift observed by HCPs describes a common theme in the literature, wherein providers felt initial hesitation with the delivery virtual care, but their perspectives favorably changed over time [19,41]. HCPs report improved satisfaction with virtual care when they receive appropriate informatics training and support and when the technology itself is user-friendly [41,42]. This information is important for future virtual care implementation and is captured in our fifth theme: considerations for implementing virtual care.

The last theme highlights that across disciplines, HCPs noted a preference for a future hybrid model of care delivery that incorporates both in-person care and virtual care. More specifically, and like other studies, providers described that assessing a patient in person for their initial appointment and then transitioning to virtual care would be the ideal model of care [19,22,43]. The importance of meeting family needs, such as patient and caregiver preferences and addressing travel and financial considerations while providing evidence-based health care, was also emphasized in this study.

The current study provides important information on HCPs’ experiences in delivering virtual pediatric MDT chronic pain care. However, there are several limitations to this study that need to be considered. The first limitation relates to the generalizability of our findings. When considering the dynamic nature of the COVID-19 pandemic, it is important to highlight that data were collected during a specific point of time during the pandemic (e.g., February–April 2021). HCPs’ perspectives on virtual care may differ in later waves of the pandemic and recovery. Future studies may consider using longitudinal methods to investigate the changing perspectives of HCPs as the pandemic has evolved. Relatedly, all participants in this study worked within an urban Canadian tertiary care setting. Our results, therefore, may not be generalizable to providers working in either rural and remote locations or alternative ambulatory settings. Furthermore, because of the nature of the methodology (questionnaire) and the small sample size, our capacity to conduct formal statistical analyses was limited, albeit still informative. Future work may consider expanding recruitment to examine experiences associated with MDT teams on a larger scale, and they could also investigate HCPs’ experiences by using other virtual care applications, such as mobile phone applications in the treatment of pediatric chronic pain [44,45].

## 5. Conclusions 

Virtual care for pediatric chronic pain has become an essential tool during the COVID-19 pandemic, and its benefits will continue to be relevant even after the pandemic has passed. In a post-COVID-19 scenario, virtual care provides accessibility, flexibility, and convenience to patients, families, and providers. Future virtual care models may look to incorporate tools such as video consultations, mobile phone applications, online educational resources, and remote monitoring devices [44,45,46]. These tools can facilitate communication between healthcare professionals and patients, allowing providers to monitor treatment progress and adjust treatment plans as needed. Virtual care also provides opportunities for caregivers to receive support and education, improving their ability to manage their child’s pain and overall wellbeing.

Another consideration is the integration of virtual care into existing pediatric pain management programs, including multidisciplinary pain clinics. A recent study by the authors examined experiences of patients and caregivers who received virtual MDT for pediatric chronic pain between March 2020 and August 2021 at the pediatric chronic pain clinic where the present study was conducted. Patients and caregivers were overall satisfied with virtual care, and the most reported preference was for a hybrid model of care incorporating at least some in-person contact with providers. In light of these findings, the chronic pain clinic at which the current study was undertaken has implemented a pilot triage system by which patients are scheduled for virtual or in-person appointments on the basis of personal and provider preferences, access to technology, and specific presentations that require in-person physical exam (e.g., Complex Regional Pain Syndrome); see [27] for further details.

Finally, recommendations and guidelines must be established for the safe and effective use of virtual care for the treatment of pediatric chronic pain. Consensus guidelines are needed to identify whether certain patient presentations are best seen in person and whether specific methodologies for assessment and treatment should be considered for provision of multidisciplinary pediatric pain care. Additionally, recommendations are required for optimizing the use of virtual care for both patients and providers (e.g., an appropriate confidential space, sufficient lighting), as well as ensuring virtual care resource accessibility for patients and providers (e.g., stable internet connection, suitable devices). Finally, specialized training should be considered for safety situations that may arise when providing virtual care (e.g., disclosures of abuse or suicidal ideation).

Overall, this study adds depth to the emerging literature capturing HCPs experiences of delivering virtual care for pediatric chronic pain throughout the pandemic. Our mixed methods study allowed for a unique understanding of these experiences within a multidisciplinary team and underscores the value of virtual care to assess and treat pediatric chronic pain. Moving forward, it will be important to incorporate these findings with other studies that conduct similar investigations throughout the pandemic to inform the creation of evidence-based educational resources and training tools for facilitation of optimal virtual care in a post-COVID world.

## Figures and Tables

**Figure 1 ijerph-20-04757-f001:**
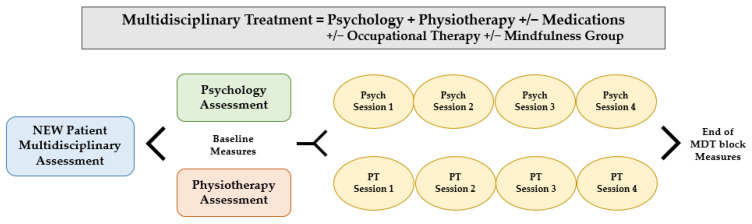
Patient flow from new intake to completion of MDT block. Psych: psychology, PT: physiotherapy, MDT: multidisciplinary.

**Table 1 ijerph-20-04757-t001:** Themes and subthemes generated from the semistructured interviews.

Theme	Subthemes
Adaptation to Virtual Care	Teamwork/togetherness Virtual modality necessitating innovations in care delivery Ease of transfer to virtual care given service model
Benefits of Virtual Care	Convenience and comfort Continuity of care
Limitations of Virtual Care	Limitations to observing nonverbal cues Barriers to physical examination Privacy concerns Technology glitches Virtual fatigue and engagement challenges Inequities in access
Shifting Stance on Virtual Care	Change in HCPs’ opinions on virtual care Change in patient and caregiver perception of virtual care
Considerations for Implementing Virtual Care	Preference for a hybrid model Recommendations for implementing virtual care

**Table 2 ijerph-20-04757-t002:** Respondent characteristics.

HCP Type	Survey (*n* = 13)	SSI (*n* = 6)
	*n* (%)	*n* (%)
Rehabilitation *	5 (38.5)	2 (33.3)
Mental Health **	3 (23.1)	2 (33.3)
Physician	3 (23.1)	1 (16.6)
Nursing	2 (15.4)	1 (16.6)
**Years of work on chronic pain team**		
<5 years	3 (23.1)	2 (33.3)
5–10 years	3 (23.1)	2 (33.3)
10–15 years	4 (30.1)	1 (16.6)
>15 years	3 (23.1)	1 (16.6)
**Experience providing virtual care**		
≤12 months	4 (30.1)	2 (33.3)
13–18 months	6 (46.2)	4 (66.7)
19–36 months	3 (23.1)	-

* Physiotherapy and occupational therapy; ** psychology and psychiatry; HCP = healthcare professional, SSI = semistructured interview.

**Table 3 ijerph-20-04757-t003:** Satisfaction survey responses: All healthcare professionals in agreement.

Likert Question	*n* (%)
I was able to develop a therapeutic relationship with the patient and caregiver in videoconference.	13 (100)
Virtual care improves my access to patients managing chronic pain.	13 (100)
I am confident that I can provide effective treatment for my patient’s needs over virtual care.	13 (100)
I had easy access to videoconferencing.	13 (100)
I was satisfied with the quality of the sound (audio).	13 (100)

**Table 4 ijerph-20-04757-t004:** Satisfaction survey responses: presented by discipline.

Likert Questions	Agree Neither * Disagree	HCP Type		
		MD (*n* = 5) *n* (%)	R (*n* = 5) *n* (%)	MH (*n* = 3) *n* (%)
I was satisfied with the quality of the picture (video).	A	4 (80)	5 (100)	3 (100)
	N	-	-	-
	D	1 (20)	-	-
I felt able to carry out effective treatment.	A	4 (80)	5 (100)	3 (100)
	N	1 (20)	-	-
	D	-	-	-
I was able to gain sufficient subjective and objective information of the patient’s presenting issues (physical/mental health) in the virtual assessment to make a diagnosis.	A	5 (100)	4 (80)	3 (100)
	N	-	1 (20)	-
	D	-	-	-
I was able to maintain patient and caregiver privacy during the virtual clinic assessment.	A	5 (100)	5 (100)	2 (67)
	N	-	-	-
	D	-	-	1 (33)
Virtual care improves the management of patients who have chronic pain.	A	4 (80)	5 (100)	3 (100)
	N	1 (20)	-	-
	D	-	-	-
Virtual care has provided new opportunities for treatment (e.g., shaping behaviors within the home environment, improving the use of technology in sessions).	A	5 (100)	4 (80)	3 (100)
	N	-	1 (20)	-
	D	-	-	-
The ability to provide treatment over virtual care has had personal benefits (e.g., ability to work from home, better time management).	A	4 (80)	4 (80)	3 (100)
	N	1 (20)	1 (20)	-
	D	-	-	-
I was confident that I could assess the patient as effectively through video as I would be if I was there in person.	A	5 (100)	4 (80)	2 (67)
	N	-	-	-
	D	-	1 (20)	1 (33)
I believe that virtual care has improved patient attendance at sessions (i.e., fewer missed appointments, greater access to parents in treatment, etc.).	A	5 (100)	3 (60)	3 (100)
	N	-	2 (40)	-
	D	-	-	-
Patients were as focused and attentive during virtual care sessions as they typically are during in-person care.	A	5 (100)	5 (100)	1 (33)
	N	-	-	-
	D	-	-	2 (67)
I experienced few/no technical difficulties during the videoconference (e.g., unexpected disconnections, loss of sound or picture).	A	5 (100)	5 (100)	2 (67)
	N	-	-	-
	D	-	-	1 (33)
It was no more challenging to discuss sensitive topics (e.g., self-injury) during virtual sessions.	A	4 (80)	2 (40)	3 (100)
	N	-	2 (40)	-
	D	1 (20)	1 (20)	-
The lack of physical contact in a virtual clinic is NOT a problem for managing chronic pain.	A	4 (80)	1 (20)	3 (100)
	N	-	1 (20)	-
	D	1 (20)	3 (60)	-

* Neither agree nor disagree. HCP = healthcare professional, MD = medicine, R = rehabilitation (physiotherapy and occupational therapy), MH = mental health (psychology and psychiatry), A = agree, N = neither agree nor disagree, D = disagree.

## Data Availability

The data presented in this study are available on request from the corresponding author. The data are not publicly available due to Institutional Research Ethics Board restrictions.

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
