# Peer review of "“Working Together”: Perspectives of Healthcare Professionals in Providing Virtual Care to Youth with Chronic Pain during the COVID-19 Pandemic"

_ijerph, 2023, doi:10.3390/ijerph20064757_

Round 1

Reviewer 1 Report

Dear Editor,

This article presents a mixed methods study on the experiences of healthcare professionals in providing multidisciplinary treatment for pediatric chronic pain through virtual care during the COVID-19 pandemic. The results from qualitative interviews and quantitative satisfaction surveys show that virtual care can be an effective model for providing appropriate diagnoses, recommendations, and care plans for pediatric chronic pain. The five themes identified from the interviews provide valuable insights into the adaptation to virtual care, benefits, limitations, a shift in stance, and considerations for implementation. Overall, the study offers a rich exploration of the experiences of healthcare professionals in providing virtual care for pediatric chronic pain. Thus it may contribute to the development of future guidelines for virtual care delivery in this population.

Minor revisions:

1. Seek to provide perspectives for use in a post-COVID scenario. In other words, focus on the goal of offering viewpoints and suggestions for how something can be utilized in a world after the COVID-19 pandemic has passed.

2. For this aim you can refer to other approaches: see doi: 10.3390/ijerph182413372.;

3. A hybrid model of care has been proposed for pain management.  10.21873/invivo.13090. these attempts should be discussed

4. Please add these references as they can add depth and context to the article, making it a more robust and valuable resource for its intended audience.

5. correct typos. E.g., lines 456-457 the acronym HCP was previously explained

A list of additional reference for consider:

doi:10.3390/ijerph20032091
doi:10.3390/ijerph20032373  doi:10.3390/ijerph20032675

doi:10.1016/j.yapd.2022.04.002

doi: 10.1089/tmj.2022.0443

doi: 10.2196/38661

Reviewer 2 Report

The treatment of chronic pain is an interesting issue that needs to be explored. In medical databases on the selected topic, there are not enough publications covering this topic.

The introduction of the work seems rather modest, lacking a presentation of the issue in the broader background.

The bibliography needs to be adjusted and unified.

The material and methods are fairly well described. But the very small group of respondents does not allow broader inference to national or global populations. In my opinion, the study should be conducted on a larger group.

Recruitment rules are not entirely clear to me - please clarify.

The results are described in words - there are no tables or synthetic presentation of the results. Too many citations. Table two is too large and unreadable - I suggest dividing the data into sections and creating additional tables/charts.

Discussion prepared correctly. Conclusions too modest, not fully related to the study conducted. They require improvement and expansion.

Reviewer 3 Report

The study aimed to examine the experiences of health care professionals (HCPs) who delivered multidisciplinary pediatric chronic pain care throughout the second wave of the Canadian COVID-19 pandemic.

In the introduction section, the authors need to mention details about Canadian Pain Task Force. Introduction section needs to be elaborated. People living with pain have already faced long waiting times for the assessment and management of their pain in these clinics. These facts need to be highlighted.

Did the subjects have access to complementary therapies?

Pain needs to be discussed especially when related to quality of life.

What may be the role of multidisciplinary pain clinics?

Could the generalizability of findings that were collected at a specific point in time influence the results?

What virtual care guidelines and policies are needed for future?

Round 2

Reviewer 2 Report

Thank you for your responses to the review, the corrections made are sufficient.

Reviewer 3 Report

All necessary corrections were made.